# Selection of timing of continuous renal replacement therapy in patients with acute kidney injury: A meta-analysis of randomized controlled trials

**Jiawei Qi[1,2], Wenwen Wu[2], Jingzhu Wang[1], Xin Guo[1,2], Chengyun Xia[1,2]***

**1** Department of Nephrology, Affiliated Hospital of North Sichuan Medical College, Nanchong, Sichuan, China, **2** Department of Clinical Medicine, North Sichuan Medical College, Nanchong, Sichuan, China

* xcy666999@sina.com

## Abstract

Acute kidney injury (AKI) is associated with high death rates and unfavorable outcomes. Previous studies evaluating the effect of the timing of CRRT therapy on the prognosis of patients with AKI have shown inconsistent results. Consequently, we aimed to assess the impact of continuous renal replacement therapy (CRRT) initiation on the outcomes of patients with AKI. This meta-analysis identified eligible randomized controlled trials (RCTs) via comprehensive searches of PubMed, Embase, and the Cochrane databases from their creation until June 1, 2024. Outcomes, including 28-, 60-, and 90-day mortality and adverse event incidence, were compared between the early and delayed CRRT groups post-randomization. Twelve RCTs (n = 1,244) were included. Meta-analysis indicated that early initiation of CRRT did not significantly affect 28-day mortality (RR 0.91 [0.79, 1.06]; p = 0.23; $I^2$ = 0). Early CRRT initiation correlated with a shorter length of ICU stay [MD -3.24 (-5.14, -1.35); p = 0.0008; $I^2$ = 36%] but did not significantly affect hospital stay duration [MD -7.00 (-14.60, 0.60); p = 0.07; $I^2$ = 38%]. The early initiation of CRRT was associated with a significant reduction in RRT dependency at discharge [RR 0.57 (0.32, 0.99); P = 0.05; $I^2$ = 0%; P = 0.47]. Compared to delayed CRRT, early CRRT was associated with higher incidence rates of hypotension [RR 1.26 (1.06, 1.50); p = 0.008; $I^2$ = 0%], thrombocytopenia [RR 1.53 (1.11, 2.10); p = 0.009; $I^2$ = 0%], and hypophosphatemia [RR 3.35 (2.18, 5.15); p < 0.00001; $I^2$ = 11%]. Our findings suggest that although early CRRT initiation is associated with short intensive care unit stays and reduced RRT dependence, it has no significant effect on mortality and is in fact associated with higher incidence rates of hypotension, thrombocytopenia, and hypophosphatemia. Therefore, early CRRT should be used clinically with caution and consideration of potential adverse effects.

## Introduction

Acute kidney injury (AKI) is associated with high mortality and unfavorable outcomes, with a rising global incidence [1–3]. Life-threatening consequences associated with AKI, such as metabolic acidosis, pulmonary edema, and hyperkalemia, are treated with renal replacement

**Data availability statement:** All relevant data are within the manuscript and its Supporting Information files.

**Funding:** The author(s) received no specific funding for this work.

**Competing interests:** The authors have declared that no competing interests exist.

therapy (RRT). However, RRT can lead to RRT-related complications and increase health-care resource utilization. Consequently, the optimal initiation timing for RRT without severe complications remains debatable. While several studies have demonstrated that early RRT initiation may offer a survival advantage for patients with AKI [4,5], recent meta-analyses [6–8] have reported conflicting results. Further, there is no comprehensive analysis including all current randomized controlled trials (RCTs) that focus on continuous renal replacement therapy (CRRT) as a treatment option for patients with AKI. While the Cochrane review does include a sub-analysis by modality, it only examines a limited number of outcome metrics and incorporates few studies related to CRRT. Therefore, in this meta-analysis, we aimed to assess the prognosis of patients with AKI by comparing the effects of early versus delayed CRRT commencement.

## Methods

The Preferred Reporting Items for Systematic Reviews and Meta-Analyses (PRISMA 2020) guidelines were followed in the reporting of this meta-analysis (**S1** Checklist).

### Search strategy and selection process

As of June 1, 2024, comprehensive searches were carried out in the PubMed, Embase, and Cochrane databases using a combination of subject terms and free words related to AKI, RRT, timing, and delay. In addition, references from relevant meta-analyses and studies were manually reviewed. **S1 Table** presents detailed search strategies. J.Q. and W.W. independently conducted study search and screening, The search strategy was developed jointly by J.Q. and W.W. resolving discrepancies through discussion.

### Criteria for inclusion and exclusion

The inclusion criteria were as follows: (1) population: patients with AKI of various causes; (2) interventions: early CRRT initiation in the intervention group; (3) comparison: early versus delayed CRRT initiation; (4) outcomes: at least one of the 28-, 60-, 90-, or 14-day mortality; and (5) study type: RCT. The exclusion criteria were as follows: (1) non-randomized studies; (2) lack of a well-defined early versus delayed initiation strategy; (3) initial treatment in the intervention group not being CRRT-related; and (4) repeated analysis of experimental data. No restrictions were placed on the publication language.

### Data gathering and bias risk evaluation

Data were extracted independently by J.Q. and W.W., including the basic characteristics of the included RCTs (authors, year and country of publication, design and setting, patient mean age, scores before randomization [SOFA or APACHE II], number of patients, male-female ratio, type of CRRT initiation), adverse events, and the primary and secondary results. Using the Cochrane risk of bias methodology (RoB 2), J.Q. and W.W. evaluated the risk of bias independently. Discrepancies were resolved through discussion or consultation with J.W.

### Outcomes

The primary outcome was 28-day mortality. Secondary outcomes were mortality at days 14, 60, and 90; ICU and hospital-based mortality; number of patients receiving RRT; number of patients dependent on RRT (at discharge and day 28); length of stay in the ICU and hospital; and mechanical ventilation days. The incidence of adverse events, including bleeding, hypotension, hypophosphatemia, arrhythmia, hypocalcemia, and thrombocytopenia, was also assessed.

## Data analysis

Analyses were carried out using RevMan 5.4. The Mantel–Haenszel method was used to calculate the risk ratios (RRs) and 95% confidence intervals (CIs) for binary variables, whereas the inverse variance method determined the mean differences (MDs) and 95% CIs for continuous variables. Heterogeneity was evaluated using the $\chi^2$ test and $I^2$ statistics. When the p-value was greater than 0.10, and $I^2$ was less than 30%, we used the fixed-effects model; in contrast, when the p-value was less than or equal to 0.10, or $I^2$ was greater than or equal to 30%, we used the random-effects model. Statistical significance was indicated by p-values less than 0.05. Funnel plots were used to evaluate publication bias. Subgroup analyses were based on baseline characteristics before randomization, including the average age of the participants (≥64 years or <64 years), cause of AKI (sepsis or other factors), and severity of the condition (SOFA score ≥12 or APACHE II score ≥25). For continuous variables, where only the median (Md) and interquartile range (IQR) were reported, we used standard approximation methods to obtain the mean (M) and standard deviation (SD) as follows: Md = M, IQR = 1.35SD. A sensitivity analysis of the primary result was also performed to evaluate the effect of single studies on overall 28-day mortality by systematically excluding individual studies and altering the effect model.

## Grading of evidence

The quality of evidence for primary and secondary outcomes, as well as adverse reactions, was graded using the GRADEpro Guideline Development Tool software.

## Results

A total of 3,575 records were obtained from the three databases by the search strategy; of these, 1108 were duplicates; nine studies were deemed eligible for inclusion after the full text and abstracts were screened; the remaining three were found through citation searching; eventually, 12 studies were included in the meta-analysis [9–20] (Fig 1). The twelve studies included 1244 patients, with 617 in the early CRRT group and 627 in the delayed group. The causes of AKI included sepsis (n = 5), post-cardiac surgery (n = 2), and multiple factors (n = 5). Study characteristics are presented in Table 1. The information for included and excluding studies after removing duplicate records are detailed in S2 Table. The quality assessment rated the studies as having a moderate or low risk of bias (S1 Fig and S3 Table). More detailed research data are available in S4 Table.

### Main outcome

The main outcome, 28-day mortality, was reported in nine studies [9–14,17–19], including 984 patients. The pooled mortality rate was 38.93% (190/488) in the early CRRT group and 42.74% (212/496) in the delayed CRRT group. Pooled analysis indicated a non-significantly lower 28-day mortality incidence in the early CRRT group (Fig 2a), with low heterogeneity between studies (p = 0.60; $I^2$ = 0%). A fixed-effects model was applied. The funnel plot did not indicate significant publication bias (S2 Fig).

### Secondary outcome

**Mortality.** Five studies assessed 60-day mortality [10,12,13,19,20] [Fig 2b] (792 patients), four examined 90-day mortality [10,12,13,19] [Fig 2c] (636 patients), three evaluated 14-day mortality [14–16] [Fig 2d] (139 patients), and four investigated in-hospital mortality [9,10,12,13] [Fig 2e] (644 patients). The analysis indicated no significant between-group differences in 60-day mortality [RR 0.91 (0.70, 1.17); P = 0.46; $I^2$ = 62%; P = 0.03], 90-day

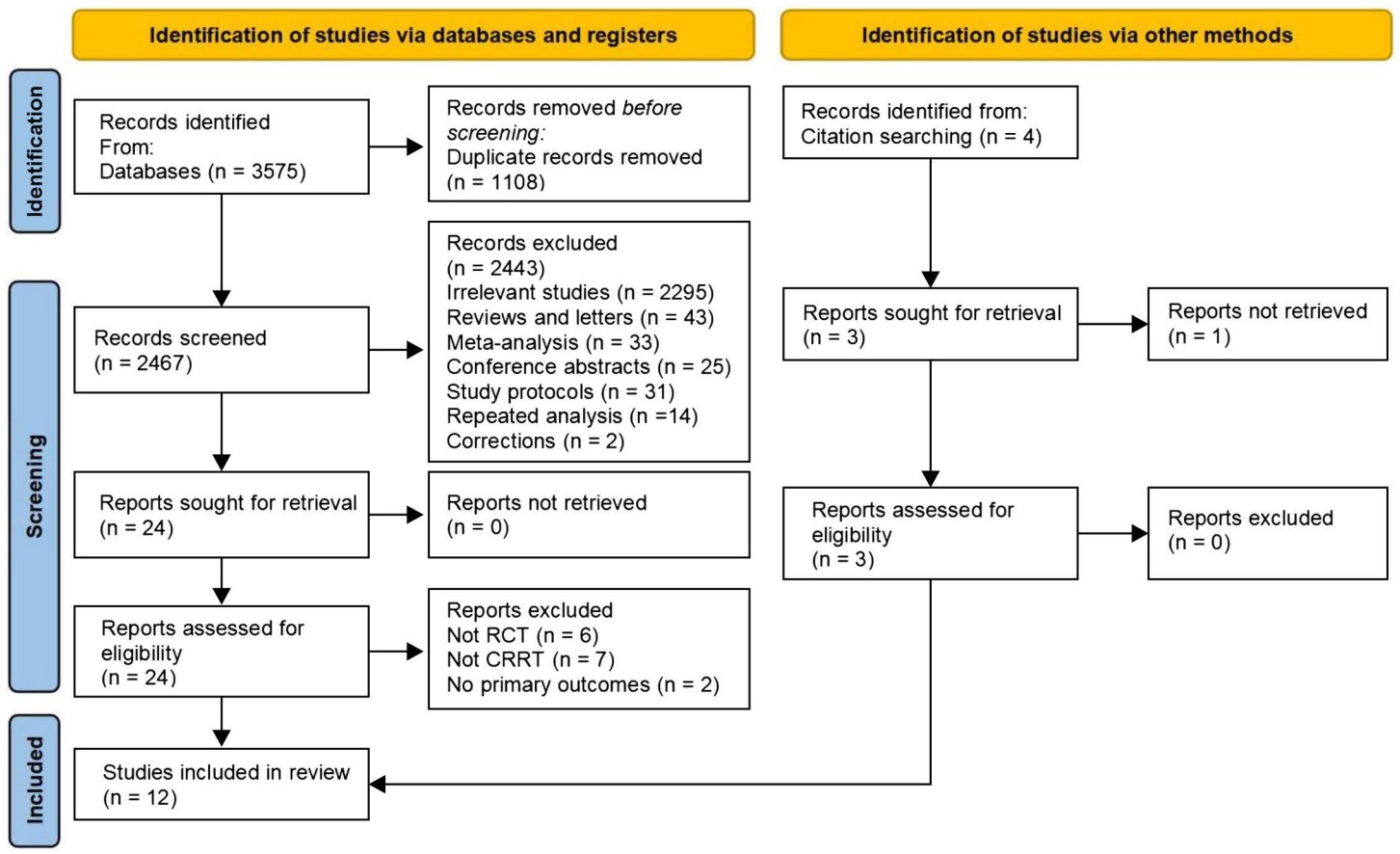

**Fig 1. PRISMA 2020 flow diagram.** RCT randomized controlled trial, CRRT continuous renal replacement therapy.

mortality [RR 1.00 (0.77, 1.28); P = 0.97; $I^2$ = 61%; P = 0.05], 14-day mortality [RR 0.73 (0.34, 1.57); P = 0.42; $I^2$ = 78%; P = 0.01], or hospital-based mortality [RR 1.03 (0.85, 1.26); P = 0.76; $I^2$ = 36%; P = 0.20]. $I^2 \geq 30\%$ or $p \leq 0.10$, all using a random-effects model.

**Number of patients receiving RRT.** Five studies reported on the number of patients receiving RRT [9–13], with 99.41% (335/337) of the patients in the early CRRT group and 73.49% (255/347) of them in the delayed CRRT group undergoing RRT during treatment. The early CRRT group exhibited a significantly higher proportion of patients receiving RRT [RR 1.41 (1.09, 1.83); P = 0.009; $I^2$ = 94%; $p \leq 0.00001$] (S3 Fig a).

**Number of patients dependent on RRT (at discharge and day 28).** Six studies [10–13,16,17] (407 patients) documented RRT dependency on day 28, and four studies [9,10,12,13] (332 patients) recorded RRT dependency at discharge. Comparative analysis revealed no significant differences in RRT dependency on day 28 [RR 0.62 (0.33, 1.18); P = 0.15; $I^2$ = 45%; P = 0.11] (S3 Fig b). However, early initiation of CRRT was associated with a significant reduction in RRT dependency at discharge [RR 0.57 (0.32, 0.99); P = 0.05; $I^2$ = 0%; P = 0.47] (S3 Fig c).

**Length of stays in the ICU and hospital.** Five studies [9,12,13,18,19] (625 patients) and four studies [9,12,13,19] (483 patients) reported the length of stay in the ICU and hospital, respectively. Meta-analysis indicated a significantly shorter ICU stay for the early CRRT group [MD -3.24 (-5.14, -1.35); P = 0.0008; $I^2$ = 36%; P = 0.18] (S3 Fig d). However, no apparent difference in length of hospital stay was observed between groups [MD -7.00 (-14.60, 0.60); P = 0.07; $I^2$ = 38%; P = 0.18] (S3 Fig e).

**Table 1. Characteristics of the included studies.**

| Study | Publication date | Country | Population included | Mean age (patients) | Scores before randomization | Sex ratio (men, women) | CRRT Mode | Outcome |
|---|---|---|---|---|---|---|---|---|
| Combes et al. [10] | 2015 | France | Cardiac surgery | 59.50 (224) | SOFA: 11.80 | 79%, 21% | HVHF, CVVHDF | 30-day mortality |
| Srisawat et al. [11] | 2018 | Thailand | Mixed | 66.80 (40) | SOFA: 9.30 | 55%, 45% | CVVH | 28-day mortality |
| Zarbock et al. [12] | 2016 | Germany | Surgical | 67.00 (231) | SOFA: 15.80 | 63%, 37% | CVVHDF | 90-day mortality |
| Lumlertgul et al. [13] | 2018 | Thailand | Mixed | 67.10 (118) | SOFA: 12 | 49%, 51% | CVVH | 28-day mortality |
| Geri et al. [14] | 2019 | France | Medical | 67.3 (35) | SOFA: 8.70 | 71%, 29% | HCOCVVHD CVVH | 28-day mortality |
| Payen et al. [15] | 2009 | France | Sepsis | 58.1 (76) | SOFA: 11.00 | 71%, 29% | CVVH | 14-day mortality |
| Sugahara et al. [16] | 2004 | Japan | Cardiac surgery | 64.50 (28) | APACHE II: 18.50 | 64%, 36% | CVVHD | 14-day mortality |
| Xia et al. [17] | 2019 | China | Sepsis | 66.40 (60) | SOFA: 9.60 | 55%, 45% | CVVH | 28-day mortality |
| Yang et al. [18] | 2019 | China | Sepsis | 58.90 (142) | NA | 61%, 39% | CVVH | 28-day mortality |
| Yin et al. [19] | 2018 | China | Sepsis | 60.80 (63) | APACHE II: 26.60 | 67%, 33% | CVVH | 28-day mortality |
| Bouman et al. [9] | 2002 | Netherland | Mixed | 68.50 (71) | SOFA: 10.40 | 59%, 41% | LVHF | 28-day mortality |
| An et al. [20] | 2021 | China | Sepsis | 60.30 (156) | APACHE II: 22.20 | 57%, 43% | CVVH | 60-day mortality |

SOFA, sepsis-related Organ Failure; APACHE II, Acute Physiology and Chronic Health Evaluation II; CRRT, continuous renal replacement therapy; RRT, renal replacement therapy; CVVH, continuous venovenous hemofiltration; CVVHD, continuous venovenous hemodialysis; CVVHDF, continuous venovenous hemodiafiltration; LVHF, low-volume hemofiltration; HVHF, high-volume hemofiltration; HCO-CVVHD, high cut-off continuous venovenous hemodialysis; NA, not available.

**Duration of mechanical ventilation.** Four studies encompassing 589 patients reported the duration of mechanical ventilation [9,10,12,19]. The analysis revealed no significant between-group difference in the duration of mechanical ventilation, including survivors and non-survivors [MD -1.67 (-4.24, 0.91); P = 0.20; $I^2$ = 75%; P = 0.008] (S3 Fig f).

**Adverse events during treatment.** Three studies involving 573 patients reported hypotension [10,12,13]. The early CRRT group experienced a higher incidence of hypotensive events [RR 1.26 (1.06, 1.50); P = 0.008; $I^2$ = 0%; P = 0.40] (Fig 3a). Similarly, three studies with 358 patients reported an increased incidence of thrombocytopenia in the early CRRT group [RR 1.53 (1.11, 2.10); P = 0.009; $I^2$ = 0%; P = 0.94] (Fig 3b). Hypophosphatemia was also more prevalent in the early CRRT group, as indicated by two studies with 342 patients [RR 3.35 (2.18, 5.15); P < 0.00001; $I^2$ = 11%; P = 0.29] (Fig 3c). For arrhythmia, hypocalcemia, and bleeding events, the RRs respectively were 1.41 [(0.83, 2.41); P = 0.21; $I^2$ = 0%; P = 0.60] (Fig 3d), 1.12 [(0.92, 1.36); P = 0.27; $I^2$ = 0%; P = 0.90] (Fig 3e), and 1.08 [(0.75, 1.56); P = 0.68; $I^2$ = 22%; P = 0.28] (Fig 3f).

## Grading of evidence

The results of evidence quality evaluation are presented in S5 Table. The quality of evidence was deemed "high" for the following outcomes: 28-day mortality, hospital mortality, and adverse events (including hypotension, thrombocytopenia, hypophosphatemia, hypocalcemia, and bleeding events).

## Subgroup and sensitivity analyses

Subgroup analysis demonstrated that variables such as average age of participants (≥64 years or <64 years), AKI etiology (sepsis or other factors), and condition severity (SOFA scores ≥ 12 or APACHE II scores ≥ 25) did not significantly affect 28-day mortality (Table 2, S4 Fig, S5 Fig, and S6 Fig). Sensitivity analysis, involving the exclusion of individual studies and adjustments to the effect model, confirmed the robustness of the primary result (Table 3 and S7 Fig).

**a 28-day mortality**

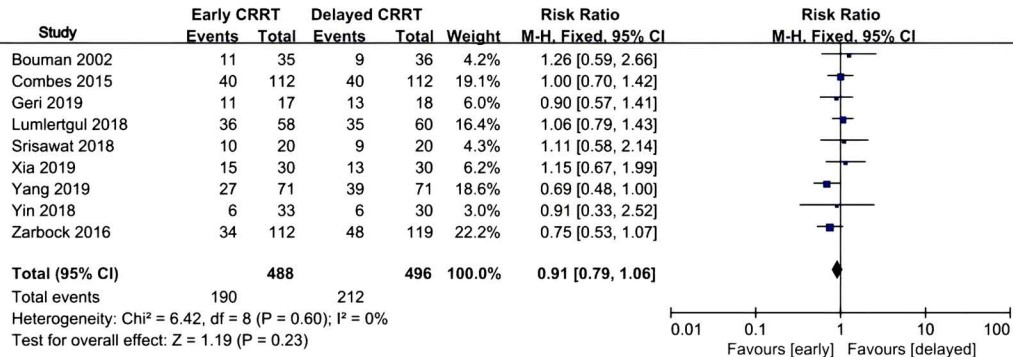

| Study | Early CRRT Events | Total | Delayed CRRT Events | Total | Weight | Risk Ratio M-H, Fixed, 95% CI |
|---|---|---|---|---|---|---|
| Bouman 2002 | 11 | 35 | 9 | 36 | 4.2% | 1.26 [0.59, 2.66] |
| Combes 2015 | 40 | 112 | 40 | 112 | 19.1% | 1.00 [0.70, 1.42] |
| Geri 2019 | 11 | 17 | 13 | 18 | 6.0% | 0.90 [0.57, 1.41] |
| Lumlertgul 2018 | 36 | 58 | 35 | 60 | 16.4% | 1.06 [0.79, 1.43] |
| Srisawat 2018 | 10 | 20 | 9 | 20 | 4.3% | 1.11 [0.58, 2.14] |
| Xia 2019 | 15 | 30 | 13 | 30 | 6.2% | 1.15 [0.67, 1.99] |
| Yang 2019 | 27 | 71 | 39 | 71 | 18.6% | 0.69 [0.48, 1.00] |
| Yin 2018 | 6 | 33 | 6 | 30 | 3.0% | 0.91 [0.33, 2.52] |
| Zarbock 2016 | 34 | 112 | 48 | 119 | 22.2% | 0.75 [0.53, 1.07] |
| | | | | | | |
| **Total (95% CI)** | | **488** | | **496** | **100.0%** | **0.91 [0.79, 1.06]** |
| Total events | 190 | | 212 | | | |

Heterogeneity: Chi² = 6.42, df = 8 (P = 0.60); I² = 0%
Test for overall effect: Z = 1.19 (P = 0.23)

**b 60-day mortality**

| Study | Early CRRT Events | Total | Delayed CRRT Events | Total | Weight | Risk Ratio M-H, Random, 95% CI |
|---|---|---|---|---|---|---|
| An 2021 | 18 | 78 | 33 | 78 | 15.6% | 0.55 [0.34, 0.88] |
| Combes 2015 | 48 | 112 | 42 | 112 | 22.5% | 1.14 [0.83, 1.58] |
| Lumlertgul 2018 | 45 | 58 | 44 | 60 | 28.5% | 1.06 [0.86, 1.30] |
| Yin 2018 | 12 | 33 | 9 | 30 | 9.4% | 1.21 [0.60, 2.46] |
| Zarbock 2016 | 43 | 112 | 60 | 119 | 23.9% | 0.76 [0.57, 1.02] |
| | | | | | | |
| **Total (95% CI)** | | **393** | | **399** | **100.0%** | **0.91 [0.70, 1.17]** |
| Total events | 166 | | 188 | | | |

Heterogeneity: Tau² = 0.05; Chi² = 10.55, df = 4 (P = 0.03); I² = 62%
Test for overall effect: Z = 0.73 (P = 0.46)

**c 90-day mortality**

| Study | Early CRRT Events | Total | Delayed CRRT Events | Total | Weight | Risk Ratio M-H, Random, 95% CI |
|---|---|---|---|---|---|---|
| Combes 2015 | 51 | 112 | 43 | 112 | 26.4% | 1.19 [0.87, 1.62] |
| Lumlertgul 2018 | 47 | 58 | 44 | 60 | 34.7% | 1.11 [0.91, 1.35] |
| Yin 2018 | 12 | 33 | 10 | 30 | 10.5% | 1.09 [0.55, 2.15] |
| Zarbock 2016 | 44 | 112 | 65 | 119 | 28.3% | 0.72 [0.54, 0.95] |
| | | | | | | |
| **Total (95% CI)** | | **315** | | **321** | **100.0%** | **1.00 [0.77, 1.28]** |
| Total events | 154 | | 162 | | | |

Heterogeneity: Tau² = 0.04; Chi² = 7.67, df = 3 (P = 0.05); I² = 61%
Test for overall effect: Z = 0.04 (P = 0.97)

**d 14-day mortality**

| Study | Early CRRT Events | Total | Delayed CRRT Events | Total | Weight | Risk Ratio M-H, Random, 95% CI |
|---|---|---|---|---|---|---|
| Geri 2019 | 11 | 17 | 13 | 18 | 40.2% | 0.90 [0.57, 1.41] |
| Payen 2009 | 20 | 37 | 17 | 39 | 39.9% | 1.24 [0.78, 1.97] |
| Sugahara 2004 | 2 | 14 | 12 | 14 | 19.8% | 0.17 [0.05, 0.61] |
| | | | | | | |
| **Total (95% CI)** | | **68** | | **71** | **100.0%** | **0.73 [0.34, 1.57]** |
| Total events | 33 | | 42 | | | |

Heterogeneity: Tau² = 0.32; Chi² = 8.93, df = 2 (P = 0.01); I² = 78%
Test for overall effect: Z = 0.81 (P = 0.42)

**e Hospital mortality**

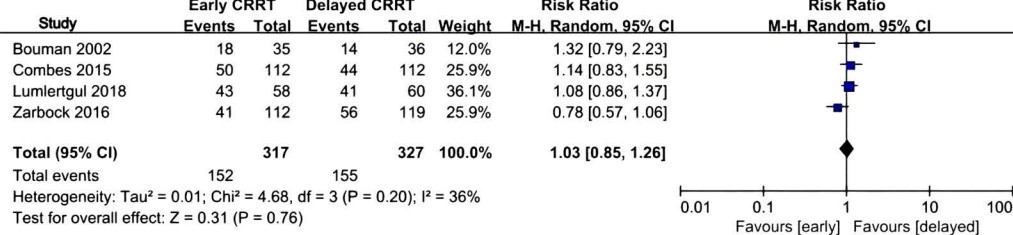

| Study | Early CRRT Events | Total | Delayed CRRT Events | Total | Weight | Risk Ratio M-H, Random, 95% CI |
|---|---|---|---|---|---|---|
| Bouman 2002 | 18 | 35 | 14 | 36 | 12.0% | 1.32 [0.79, 2.23] |
| Combes 2015 | 50 | 112 | 44 | 112 | 25.9% | 1.14 [0.83, 1.55] |
| Lumlertgul 2018 | 43 | 58 | 41 | 60 | 36.1% | 1.08 [0.86, 1.37] |
| Zarbock 2016 | 41 | 112 | 56 | 119 | 25.9% | 0.78 [0.57, 1.06] |
| | | | | | | |
| **Total (95% CI)** | | **317** | | **327** | **100.0%** | **1.03 [0.85, 1.26]** |
| Total events | 152 | | 155 | | | |

Heterogeneity: Tau² = 0.01; Chi² = 4.68, df = 3 (P = 0.20); I² = 36%
Test for overall effect: Z = 0.31 (P = 0.76)

**Fig 2. Forest plot of mortality.** CRRT, continuous renal replacement therapy; M-H, Mantel–Haenszel; CI confidence interval.

**a Hypotension**

| Study | Early CRRT Events | Total | Delayed CRRT Events | Total | Weight | Risk Ratio M-H, Fixed, 95% CI |
|---|---|---|---|---|---|---|
| Combes 2015 | 87 | 112 | 74 | 112 | 85.3% | 1.18 [1.00, 1.39] |
| Lumlertgul 2018 | 20 | 58 | 12 | 60 | 13.6% | 1.72 [0.93, 3.20] |
| Zarbock 2016 | 2 | 112 | 1 | 119 | 1.1% | 2.13 [0.20, 23.11] |
| **Total (95% CI)** | | **282** | | **291** | **100.0%** | **1.26 [1.06, 1.50]** |
| Total events | 109 | | 87 | | | |

Heterogeneity: Chi² = 1.85, df = 2 (P = 0.40); I² = 0%
Test for overall effect: Z = 2.65 (P = 0.008)

Favours [experimental]   Favours [control]

**b Thrombocytopenia**

| Study | Early CRRT Events | Total | Delayed CRRT Events | Total | Weight | Risk Ratio M-H, Fixed, 95% CI |
|---|---|---|---|---|---|---|
| Bouman 2002 | 3 | 35 | 2 | 36 | 5.0% | 1.54 [0.27, 8.68] |
| Combes 2015 | 56 | 112 | 37 | 112 | 93.7% | 1.51 [1.10, 2.09] |
| Yin 2018 | 1 | 33 | 0 | 30 | 1.3% | 2.74 [0.12, 64.69] |
| **Total (95% CI)** | | **180** | | **178** | **100.0%** | **1.53 [1.11, 2.10]** |
| Total events | 60 | | 39 | | | |

Heterogeneity: Chi² = 0.13, df = 2 (P = 0.94); I² = 0%
Test for overall effect: Z = 2.63 (P = 0.009)

Favours [early]   Favours [delayed]

**c Hypophosphatemia**

| Study | Early CRRT Events | Total | Delayed CRRT Events | Total | Weight | Risk Ratio M-H, Fixed, 95% CI |
|---|---|---|---|---|---|---|
| Combes 2015 | 57 | 112 | 19 | 112 | 90.6% | 3.00 [1.92, 4.70] |
| Lumlertgul 2018 | 13 | 58 | 2 | 60 | 9.4% | 6.72 [1.59, 28.50] |
| **Total (95% CI)** | | **170** | | **172** | **100.0%** | **3.35 [2.18, 5.15]** |
| Total events | 70 | | 21 | | | |

Heterogeneity: Chi² = 1.13, df = 1 (P = 0.29); I² = 11%
Test for overall effect: Z = 5.51 (P < 0.00001)

Favours [early]   Favours [delayed]

**d Arrhythmias**

| Study | Early CRRT Events | Total | Delayed CRRT Events | Total | Weight | Risk Ratio M-H, Fixed, 95% CI |
|---|---|---|---|---|---|---|
| Lumlertgul 2018 | 21 | 58 | 16 | 60 | 97.0% | 1.36 [0.79, 2.33] |
| Zarbock 2016 | 1 | 112 | 0 | 119 | 3.0% | 3.19 [0.13, 77.40] |
| **Total (95% CI)** | | **170** | | **179** | **100.0%** | **1.41 [0.83, 2.41]** |
| Total events | 22 | | 16 | | | |

Heterogeneity: Chi² = 0.27, df = 1 (P = 0.60); I² = 0%
Test for overall effect: Z = 1.27 (P = 0.21)

Favours [early]   Favours [delayed]

**e Hypocalcemia**

| Study | Early CRRT Events | Total | Delayed CRRT Events | Total | Weight | Risk Ratio M-H, Fixed, 95% CI |
|---|---|---|---|---|---|---|
| Lumlertgul 2018 | 4 | 58 | 4 | 60 | 5.4% | 1.03 [0.27, 3.94] |
| Zarbock 2016 | 75 | 112 | 71 | 119 | 94.6% | 1.12 [0.92, 1.37] |
| **Total (95% CI)** | | **170** | | **179** | **100.0%** | **1.12 [0.92, 1.36]** |
| Total events | 79 | | 75 | | | |

Heterogeneity: Chi² = 0.01, df = 1 (P = 0.90); I² = 0%
Test for overall effect: Z = 1.09 (P = 0.27)

Favours [early]   Favours [delayed]

**f Bleeding events**

| Study | Early CRRT Events | Total | Delayed CRRT Events | Total | Weight | Risk Ratio M-H, Fixed, 95% CI |
|---|---|---|---|---|---|---|
| Bouman 2002 | 7 | 35 | 3 | 36 | 7.4% | 2.40 [0.67, 8.55] |
| Combes 2015 | 35 | 112 | 34 | 112 | 85.2% | 1.03 [0.70, 1.52] |
| Lumlertgul 2018 | 1 | 58 | 3 | 60 | 7.4% | 0.34 [0.04, 3.22] |
| **Total (95% CI)** | | **205** | | **208** | **100.0%** | **1.08 [0.75, 1.56]** |
| Total events | 43 | | 40 | | | |

Heterogeneity: Chi² = 2.58, df = 2 (P = 0.28); I² = 22%
Test for overall effect: Z = 0.41 (P = 0.68)

Favours [experimental]   Favours [control]

**Fig 3. Forest plot of adverse events.** CRRT, continuous renal replacement therapy; M-H, Mantel–Haenszel; CI, confidence interval.

**Table 2. Subgroup analysis of 28-day mortality.**

| Group | No. of trials | No. of patients | Risk ratio (95% CI) | P-value | Heterogeneity | |
|---|---|---|---|---|---|---|
| | | | | | I² index (%) | P-value |
| **Average age** | | | | | | |
| <64 years | 3 | 429 | 0.85 (0.67, 1.09) | 0.20 | 3 | 0.36 |
| >64 years | 6 | 555 | 0.96 (0.80, 1.15) | 0.64 | 0 | 0.62 |
| **AKI etiology** | | | | | | |
| Sepsis | 3 | 265 | 0.82 (0.61, 1.09) | 0.18 | 16 | 0.30 |
| Non-sepsis/mixed factors | 6 | 719 | 0.95 (0.80, 1.13) | 0.57 | 0 | 0.68 |
| **SOFA/APACHE II scores** | | | | | | |
| Low score | 5 | 430 | 1.05 (0.83, 1.32) | 0.69 | 0 | 0.92 |
| High score | 3 | 412 | 0.89 (0.71, 1.11) | 0.30 | 13 | 0.32 |

SOFA, sepsis-related organ failure; APACHE II, Acute Physiology and Chronic Health Evaluation II; AKI, acute kidney injury.

**Table 3. Sensitivity analysis of 28-day mortality.**

| Excluded research | No. of patients | Risk ratio (95% CI) | P-value | Heterogeneity | |
|---|---|---|---|---|---|
| I² index (%) | P-value | | | | |
| Bouman 2002 | 913 | 0.90 (0.77, 1.04) | 0.17 | 0 | 0.57 |
| Combes 2015 | 760 | 0.89 (0.76, 1.05) | 0.18 | 0 | 0.52 |
| Geri 2019 | 946 | 0.92 (0.79, 1.07) | 0.26 | 0 | 0.49 |
| Lumlertgul 2018 | 866 | 0.89 (0.75, 1.05) | 0.15 | 0 | 0.63 |
| Srisawat 2018 | 944 | 0.91 (0.78, 1.05) | 0.20 | 0 | 0.53 |
| Xia 2019 | 924 | 0.90 (0.77, 1.05) | 0.17 | 0 | 0.57 |
| Yang 2019 | 842 | 0.97 (0.82, 1.13) | 0.67 | 0 | 0.83 |
| Yin 2018 | 921 | 0.91 (0.79, 1.06) | 0.24 | 0 | 0.49 |
| Zarbock 2016 | 753 | 0.96 (0.82, 1.13) | 0.63 | 0 | 0.68 |

CI, confidence interval.

## Discussion

This meta-analysis, encompassing 12 RCTs, assessed the effects of starting CRRT early versus late on AKI patient outcomes. The analysis revealed no notable variations in mortality at days 14, 28, 60, and 90 based on CRRT initiation timing, indicating no apparent survival benefit for early CRRT initiation in patients with AKI. Sensitivity analyses, including model variations and exclusion of individual RCTs, confirmed the robustness of these primary results. These results contribute to our understanding whether the CRRT modality influences outcomes in this contentious area.

Previously, the connection between the timing and results of AKI RRT has primarily been examined through retrospective studies and meta-analyses. However, recent advancements in large-scale, high-quality RCTs [21] and meta-analyses [6–8] have led to divergent conclusions, with a growing consensus that early RRT initiation does not confer a survival advantage for patients with AKI. Some meta-analyses [6–8] did not restrict the early RRT intervention mode. Li [22] limited RRT intervention to patients with sepsis, whereas Xia [23] included fewer RCTs and lacked subsequent studies. Our meta-analysis, which included all AKI patients with early CRRT initiation as the regimen, and incorporated the latest RCTs, provides a comprehensive and updated perspective.

For the interventions in the included studies, the selected CRRT modes varied, and patient criteria for transitioning to intermittent dialysis also differed. Disparities existed in the causes of AKI, CRRT initiation criteria, and inclusion criteria across the RCTs; partial pooled analyses revealed significant heterogeneity. Consequently, the results should be approached with caution.

In this study, the early CRRT group had more patients receiving RRT, compared to the delayed CRRT group, consistent with other studies [6–8]. In the delayed CRRT group, most patients, except for those who died or had other complications, did not require RRT because of renal function recovery. This indicates that implementing delayed strategies could reduce medical resource utilization and mitigate CRRT-related adverse effects. However, the AKIKI2 trial [24] randomly assigned patients with AKI to delayed and more-delayed CRRT strategy groups, with the more delayed strategy group demonstrating no additional benefits or potential harm. Thus, an appropriately delayed initiation strategy groups, tailored to the patient's specific conditions, may be more advantageous than delaying until mandatory indications such as evident hyperkalemia, acidosis, or pulmonary edema emerge.

Additionally, this meta-analysis demonstrated that the early start of CRRT significantly reduced the number of patients dependent on RRT at discharge, and no discernible difference in hospital stay was observed, except for reduced ICU stay duration. Pooled analysis of adverse events revealed a greater risk of hypotension, thrombocytopenia, and hypophosphatemia in the early CRRT group, while no significant differences were noted in cardiac arrhythmias, hypocalcemia, and bleeding events.

Early initiation of CRRT facilitates the removal of toxins and inflammatory mediators in sepsis, potentially preventing its progression. However, subgroup analysis indicated that early CRRT initiation did not confer significant benefits to patients with AKI in the sepsis subgroup. This finding aligns with that of the IDEAL-ICU trial [25], where early RRT initiation in septic patients with AKI did not result in a lower 90-day mortality rate. Although theoretically, reducing unbound cytokines might mitigate organ damage in patients with sepsis and lower mortality, cytokine aggregation in the tissue gap may influence outcomes [26]. Therefore, an optimized strategy for clearing cytokines in tissues is necessary to enhance outcomes in patients with sepsis.

This study had several strengths. First, it incorporated a greater number of RCTs using CRRT as the treatment regimen for early intervention, including the latest research from 2019 onwards [14,17,18,20]. Second, it provided a comprehensive evaluation of the impact of CRRT initiation timing on treatment results, such as mortality rates at various intervals and the incidence of adverse events. However, this study also had some limitations. Primarily, there was significant variability in CRRT initiation timing among the included studies. Additionally, the lack of subgroup analyses based on different CRRT initiation timings precludes a more nuanced assessment of the effect of CRRT on survival outcomes. Finally, some studies involved patients initially receiving CRRT but transitioning to intermittent RRT once hemodynamic stability and other requirements had been achieved.

## Conclusion

In conclusion, our findings indicate that early CRRT may increase the risk adverse outcomes without improving prognosis. Therefore, patients with AKI and those treated in settings with limited medical resources may benefit more from appropriate delays in starting CRRT.

## Supporting information

**S1 Fig. Risk of bias summary.**
(TIF)

**S2 Fig. Funnel plot to evaluate publication bias.**
(TIF)

**S3 Fig. Forest plot of the secondary outcomes.**
(TIF)

**S4 Fig. Subgroup analyses by average age of participants.**
(TIF)

**S5 Fig. Subgroup analyses by AKI etiology.**
(TIF)

**S6 Fig. Subgroup analyses by the scores.**
(TIF)

**S7 Fig. Forest plot by random effects.**
(TIF)

**S1 Checklist. PRISMA 2020 checklist.**
(DOCX)

**S1 Table. Search strategy.**
(DOCX)

**S2 Table. Studies included and excluded.**
(DOCX)

**S3 Table. Risk of bias of included studies.**
(DOCX)

**S4 Table. Raw data used in the current meta-analysis.**
(DOCX)

**S5 Table. Assessment of certainty of evidence.**
(DOCX)

## Author contributions

**Conceptualization:** Jiawei Qi, Wenwen Wu.

**Data curation:** Jingzhu Wang, Chengyun Xia.

**Formal analysis:** Jiawei Qi, Wenwen Wu.

**Investigation:** Jiawei Qi, Wenwen Wu.

**Methodology:** Jiawei Qi, Wenwen Wu, Chengyun Xia.

**Resources:** Jiawei Qi, Wenwen Wu.

**Software:** Jiawei Qi, Wenwen Wu.

**Validation:** Jiawei Qi, Wenwen Wu.

**Writing – original draft:** Jiawei Qi, Wenwen Wu, Xin Guo.

**Writing – review & editing:** Jingzhu Wang, Chengyun Xia.

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
