## [Decision Letter · Decision Letter 0]

2 Oct 2024

PONE-D-24-38687Selection of timing of continuous renal replacement therapy in patients with acute kidney injury: a meta-analysis of randomized controlled trialsPLOS ONE

Dear Dr. Xia,

Thank you for submitting your manuscript to PLOS ONE. After careful consideration, we feel that it has merit but does not fully meet PLOS ONE’s publication criteria as it currently stands. Therefore, we invite you to submit a revised version of the manuscript that addresses the points raised during the review process.

We look forward to receiving your revised manuscript.

Kind regards,

Wangari Waweru-Siika, FRCA

Academic Editor

PLOS ONE

Journal Requirements:

3. Please include captions for your Supporting Information files at the end of your manuscript, and update any in-text citations to match accordingly. Please see our Supporting Information guidelines for more information: http://journals.plos.org/plosone/s/supporting-information .

4. As required by our policy on Data Availability, please ensure your manuscript or supplementary information includes the following:

Reviewers' comments:

Reviewer's Responses to Questions

**Comments to the Author**

1. Is the manuscript technically sound, and do the data support the conclusions?

Reviewer #1: Partly

Reviewer #2: Yes

2. Has the statistical analysis been performed appropriately and rigorously? 

Reviewer #1: Yes

Reviewer #2: No

3. Have the authors made all data underlying the findings in their manuscript fully available?

Reviewer #1: Yes

Reviewer #2: Yes

4. Is the manuscript presented in an intelligible fashion and written in standard English?

Reviewer #1: Yes

Reviewer #2: Yes

5. Review Comments to the Author

Reviewer #1: Thank you for this interesting submission and for the opportunity to review.

Briefly, the authors undertook a systematic review and meta-analysis of early vs. delayed CRRT for patients with acute kidney injury. They included 12 trials and pooled outcomes suggest that early CRRT did not confer a mortality advantage while at the same increasing adverse events.

I have a number of major and minor comments for the authors to consider.

Major:

1. My biggest comment relates to the rationale or justification for undertaking this review given the rich data available in previous and relatively recent high-quality reviews (see: https://pubmed.ncbi.nlm.nih.gov/33407756/
https://pubmed.ncbi.nlm.nih.gov/33407756/ and https://pubmed.ncbi.nlm.nih.gov/32334654/ as examples). If the rationale is that these reviews were not exclusively restricted to CRRT, it still begs the question as to why we think a special comparison of early vs. late CRRT is relevant to the reader. We know from prior studies and meta-analyses that the mode of dialysis in AKI has not shown to influence outcomes (for example: https://www.sciencedirect.com/science/article/abs/pii/S0883944117301727) nor has the timing. Also, the Cochrane review includes a sub-analysis by mode. I am thus hard-pressed to understand what new information the current review is contributing.

The authors need to substantiate and provide a convincing rationale for undertaking this exercise.

2. Was the protocol registered on PROSPERO or an equivalent publicly available repository to evaluate the adherence to prespecified aims and analysis plans?

3. The authors mention they conducted the review in accordance with PRISMA guidelines (methods, lines 48 and 49): PRISMA is a guideline for reporting systematic reviews and not for the conduct. Please correct this.

Minor:

1. How were disagreements resolved? (search strategy, lines 55 and 56)

2. Exclusion criteria: what do they mean by 'repeated analysis of experimental data'? (lines 63 and 64)

3. Table 1: There seem to be issues with referencing and it shows up in the document as 'errors'. Please fix this.

4. Attention needs to be paid to consistency of decimal points throughout the manuscript. Authors should stick to 1 or 2 decimal places and be consistent.

5. Lines 111: change 'main ending' to 'main outcome' or 'primary outcome'

6. Line 113: change 'composite mortality' to 'pooled mortality'

7. Authors refer to 14-day, 60-day and 90-day mortality outcomes as rates. These are not really rates and are instead percentages or proportions. Please correct this.

8. Table 2 is best presented as a forest plot. Also what is the rationale for the break up of the age subgroup as less than 64 and more than 64 years? And how was the cut off for SOFA and APACHE defined as low and high?

9. in the PRISMA flow diagram, under 'screening' the authors report that they excluded 2443 studies- why? Please list the reasons.

Reviewer #2: The threshold value of I2 should not be used to determine random or fix models. The two methods have different assumptions and research aims. Need to evaluate more carefully the nature of studies and check whether they share a common effect.

Table 1. Study column: Error. Reference source not found.

Figure 1 provide reasons for excluding 2443 records.

Forest plot figures, Change the left column title as “Study” instead of “Study or Subgroup”.

Capitalize the first letters of some studies.

6. PLOS authors have the option to publish the peer review history of their article (what does this mean? ). If published, this will include your full peer review and any attached files.

**Do you want your identity to be public for this peer review?** For information about this choice, including consent withdrawal, please see our Privacy Policy .

Reviewer #1: No

Reviewer #2: No

---

## [Author Response · Author response to Decision Letter 0]

8 Nov 2024

Dear Reviewers and Academic Editor,

We would like to express our sincere gratitude for your time and effort in reviewing our manuscript. Your insightful comments have greatly contributed to improving our work. We have carefully addressed each comment, and the corresponding revisions are highlighted in blue in the manuscript. Our responses to your comments are also provided in blue.

We look forward to your feedback and to advancing this manuscript toward publication in PLOS ONE.

Sincerely,

Xia Chengyun

Corresponding Author

Email: xcy666999@sina.com

---

## [Decision Letter · Decision Letter 1]

22 Nov 2024

PONE-D-24-38687R1

Selection of timing of continuous renal replacement therapy in patients with acute kidney injury: A meta-analysis of randomized controlled trials

PLOS ONE

Dear Dr. Xia,

Thank you for submitting your manuscript to PLOS ONE. After careful consideration, we have decided that your manuscript does not meet our criteria for publication and must therefore be rejected.

I am sorry that we cannot be more positive on this occasion, but hope that you appreciate the reasons for this decision.

Kind regards,

Wangari Waweru-Siika, FRCA

Academic Editor

PLOS ONE

Reviewers' comments:

Reviewer's Responses to Questions

**Comments to the Author**

1. If the authors have adequately addressed your comments raised in a previous round of review and you feel that this manuscript is now acceptable for publication, you may indicate that here to bypass the “Comments to the Author” section, enter your conflict of interest statement in the “Confidential to Editor” section, and submit your "Accept" recommendation.

Reviewer #1: (No Response)

Reviewer #2: All comments have been addressed

2. Is the manuscript technically sound, and do the data support the conclusions?

Reviewer #1: Partly

Reviewer #2: (No Response)

3. Has the statistical analysis been performed appropriately and rigorously? 

Reviewer #1: Yes

Reviewer #2: (No Response)

4. Have the authors made all data underlying the findings in their manuscript fully available?

Reviewer #1: Yes

Reviewer #2: (No Response)

5. Is the manuscript presented in an intelligible fashion and written in standard English?

Reviewer #1: Yes

Reviewer #2: (No Response)

6. Review Comments to the Author

Reviewer #1: Unfortunately, my major concern related to the rationale of undertaking this exercise remains unanswered.

Also, I do not see a point by point response letter to the comments.

Reviewer #2: All my concerns were addressed.

7. PLOS authors have the option to publish the peer review history of their article (what does this mean? ). If published, this will include your full peer review and any attached files.

**Do you want your identity to be public for this peer review?** For information about this choice, including consent withdrawal, please see our Privacy Policy .

Reviewer #1: No

Reviewer #2: No

- - - - -

---

## [Author Response · Author response to Decision Letter 1]

6 Jan 2025

Response to Reviewers uploaded in Attach Files section.

---

## [Editor Report · Decision Letter 2]

18 Feb 2025

Selection of timing of continuous renal replacement therapy in patients with acute kidney injury: A meta-analysis of randomized controlled trials

PONE-D-24-38687R2

Dear Dr. Xia,

We’re pleased to inform you that your manuscript has been judged scientifically suitable for publication and will be formally accepted for publication once it meets all outstanding technical requirements.

Kind regards,

Fabio Sallustio, PhD

Academic Editor

PLOS ONE
---

## [Editor Report · Acceptance letter]

PONE-D-24-38687R2

PLOS ONE

Dear Dr. Xia,

I'm pleased to inform you that your manuscript has been deemed suitable for publication in PLOS ONE. Congratulations! Your manuscript is now being handed over to our production team.

Kind regards,

on behalf of

Prof. Fabio Sallustio

Academic Editor

PLOS ONE